# Coarse-Grained Modeling of the SARS-CoV-2 Spike Glycoprotein by Physics-Informed Machine Learning

**David Liang** [1,*]**, Ziji Zhang** [2] **, Miriam Rafailovich** [3]**, Marcia Simon** [4]**, Yuefan Deng** [2] **and Peng Zhang** [2,*]

[1] Department of Chemistry, University of Chicago, Chicago, IL 60637, USA
[2] Departments of Applied Mathematics and Statistics, Stony Brook University, Stony Brook, NY 11790, USA
[3] Materials Science and Chemical Engineering, Stony Brook University, Stony Brook, NY 11790, USA
[4] Oral Biology and Pathology, Stony Brook University, Stony Brook, NY 11790, USA
[*] Correspondence: dliang7234@uchicago.edu (D.L.); peng.zhang@stonybrook.edu (P.Z.);
Tel.: +1-(631)-377-8211 (D.L.)

**Abstract:** Coarse-grained (CG) modeling has defined a well-established approach to accessing greater space and time scales inaccessible to the computationally expensive all-atomic (AA) molecular dynamics (MD) simulations. Popular methods of CG follow a bottom-up architecture to match properties of fine-grained or experimental data whose development is a daunting challenge for requiring the derivation of a new set of parameters in potential calculation. We proposed a novel physics-informed machine learning (PIML) framework for a CG model and applied it, as a verification, for modeling the SARS-CoV-2 spike glycoprotein. The PIML in the proposed framework employs a force-matching scheme with which we determined the force-field parameters. Our PIML framework defines its trainable parameters as the CG force-field parameters and predicts the instantaneous forces on each CG bead, learning the force field parameters to best match the predicted forces with the reference forces. Using the learned interaction parameters, CGMD validation simulations reach the microsecond time scale with stability, at a simulation speed 40,000 times faster than the conventional AAMD. Compared with the traditional iterative approach, our framework matches the AA reference structure with better accuracy. The improved efficiency enhances the timeliness of research and development in producing long-term simulations of SARS-CoV-2 and opens avenues to help illuminate protein mechanisms and predict its environmental changes.

**Keywords:** coarse-grained modeling; SARS-CoV-2; molecular dynamics; machine learning

## 1. Introduction

All-atomic molecular dynamics (AAMD) simulations have defined a foundational basis for molecular modeling, providing both atomic- and femtosecond-level resolutions into the dynamic evolution of systems. However, its computational cost often limits its practical and large-scale applications beyond microsecond simulations of millions of atoms. Multiscale coarse-grained (CG) modeling defines a well-established approach within literature for simulating complex, high-definition systems using simplified, lower-resolution representations, often by aggregating groups of atoms into a single CGMD "bead," thus increasing computational efficiency [1–4]. Popular methods of CG strive to match structure properties or energy distributions of fine-grained or experimental data which center around describing a new force field, consisting of the system's parameters and potential calculations, to reproduce the properties of all-atomic (AA) reference simulations. In practice, the CG approaches do not aim to fully reproduce the distributions of the reference data, instead focusing on optimizing, and thus sacrificing complexity in favor of accessing more relevant simulation spatial and temporal scales. The optimization of an accurate and consistent CG model remains an active and significant challenge in the field [5–7].

Recent advances in machine learning (ML) have proven their strength to accelerate both in vitro and in silico biological studies [8–11]. In this work, we develop a novel physics-informed machine learning (PIML) framework for parameterization and optimization of CG force fields, resulting in the development of physics-informed CG models from fine-grained molecular dynamics (MD) to enable simulation across greater spatial and temporal scales that are inaccessible to conventional AAMD simulations. As an example, we focus on the SARS-CoV-2 spike glycoprotein in practical application. The outbreak of SARS-CoV-2 in 2019 and its continued persistence have led to millions of deaths globally [12], prompting investigations of its molecular structure and mechanisms of infection. The outer surface of the virion is covered by numerous unique spike proteins, largely responsible for the binding of the virus to the host cell receptor angiotensin-converting enzyme 2, thus mediating cell entry [13]. Hence, understanding this protein is crucial to investigating the infectivity of the virus and taking steps toward better therapeutics and vaccines. In this study, the protein serves as a prime example of both a timely and significant application for our proposed methodology. While studies are currently underway in uncovering specific mechanisms of action of the SARS-CoV-2 virion or possible therapeutics [14,15], many practical and large-scale applications of AAMD simulations are challenged by the computational expense when dealing with this S-protein of over twenty thousand atoms [16].

Efforts have been made to develop CG models with the corresponding force fields to simulate the S-protein; for instance, a hetero-elastic network model [17,18] was used to optimize bonded energy calculations, while relative-entropy minimization was applied to learn nonbonded interactions and an empirical approach was taken to refine the CG model [18]. Another study [2] utilized the iterative Boltzmann inversion method (IBIM) to reproduce the reference atomic fluctuations. We propose a novel ML-based parameterization approach that goes beyond the existing approaches by defining physics-informed force field parameters and learning the CG free energy functions that account for the entire network of bonded and nonbonded interactions. We unify the optimization task for the CG force field for more efficient parameter determination. The model, trained by a force-matching scheme, corroborates the CG forces and associated effective potential with the AAMD simulation data. This approach, offering an easily generalizable means of parametrization to different proteins and applications, differentiates from other schemes that rely on empirical or user-defined parameters.

While there exist ML-based force fields in other studies, most notably CGNet [19], and its variants CGSchNet [20], as well as TorchMD [21], they are different from our approach. While they were developed for application on smaller proteins such as alanine dipeptide or chignolin, this study aims to tackle a more challenging application with a significantly larger protein, and hence we rely on ML to derive and parameterize a force field.

The interactions of the bottom-up CG model, in our approach, use a combination of iterative and PIML strategies. The AAMD simulations, producing the ground truth, are conducted on powerful supercomputers to help obtain massive data to derive the associated CG model. Our main contributions are:

- An innovative application of supervised ML is proposed to derive a physics-informed CG model.
- The supervised ML is combined with molecular dynamics towards greater efficiency, achieving a speed-up of CGMD simulations of 40,000 over the conventional AAMD simulations while retaining structural accuracy.
- The greater efficiency enhances the timeliness of the research in producing long-term simulations and blazes a path for new applications and further investigation, i.e., protein binding and prediction of environmental changes.

The remainder of this paper is organized as follows. Section 2 describes the physics-informed CG model and its implementation. Section 3 reports the experimental results of the CGMD simulations and corroborates them with the AAMD simulations. Section 4 provides discussions and future direction in multiscale modeling of biomolecular systems.

## 2. Materials and Methods

### 2.1. Coarse-Grained Structure

The full SARS-CoV-2 S-protein model was obtained from the protein data bank 6VXX and was run through NAMD software [22,23] on the AA system consisting of 22,815 atoms (a total of 45,153 atoms including the hydrogens). The coarse-grain structure follows the established aggressive Shape-Based Coarse Graining (SBCG) approach [6], which reduced the model to 60 representing particles, maintaining the homotrimeric structure with 20 atoms per chain (Figure 1 and Table 1). Atoms were assigned to beads based on the overall topology of the macromolecule. This involved the use of a topology-preserving neural network, where each CG bead corresponds to a node in the network and the coordinates of the atoms are inputted to adapt the neural network [24]. The hyperparameters used in the SBCG GUI are as follows: initial eps = 0.3, final eps = 0.05, initial lambda = 5.0, and final lambda = 0.01, with bonds formed from the all-atom structure. Beads are uncharged, but the CG model is fitted to reproduce the electrostatics present in the AAMD simulations.

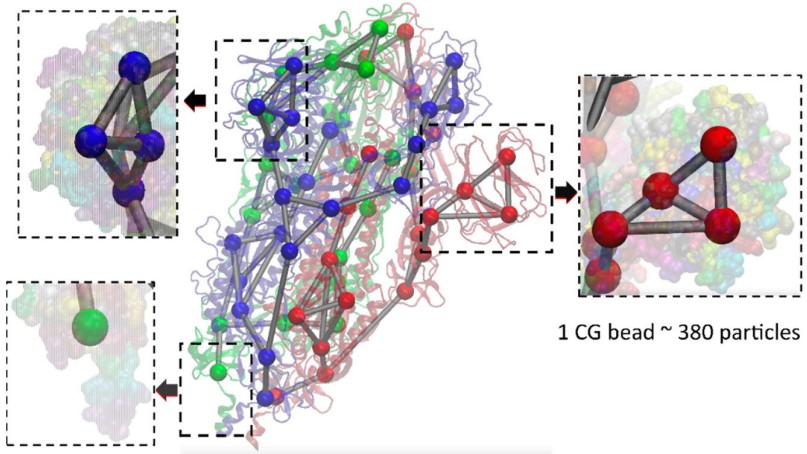

1 CG bead ~ 380 particles

**Figure 1.** Structural visualization for AA vs. CG model. Red, blue, and green denote 3 chains.

**Table 1.** Statistics for AA vs. CG model.

|           | AA Model                     | CG Model |
|-----------|------------------------------|----------|
| Atoms     | 22,815 (45,153 w/hydrogens)  | 60       |
| Bonds     | 23,385                       | 81       |
| Angles    | 31,887                       | 159      |
| Dihedrals | 37,872                       | 231      |

### 2.2. Coarse-Grained Force Field

Our physics-informed CG modeling follows a multiscale approach, characterized roughly by the transfer of high resolution AA data to the CG scale through the parameterization of a CG model [25]. The approach is shown in Figure 2. In the first box of "Data Collection," spatial and temporal mapping schemes are employed to map the AAMD simulations to the reduced-resolution CG structure, representing the ground truth. In the second box of "Parameter Optimization," a new CG force field is parameterized to conform to this ground truth. This is carried out by first employing IBIM on the bonded parameters [26,27], which iteratively scales parameters and simulates trials to match the reference radial distribution function (RDF). Visual Molecular Dynamics (VMD) software is used to initialize the non-bonded terms [28] based on approximated values and solvent-accessible-surface area (SASA) calculations [29]. In the last box of "Validation Analysis," the learned parameters are implemented in a CGMD to corroborate the proposed method

with the baseline reference. The simulations are evaluated in terms of simulation accuracy and computation speed. Specific details are provided in the following sections.

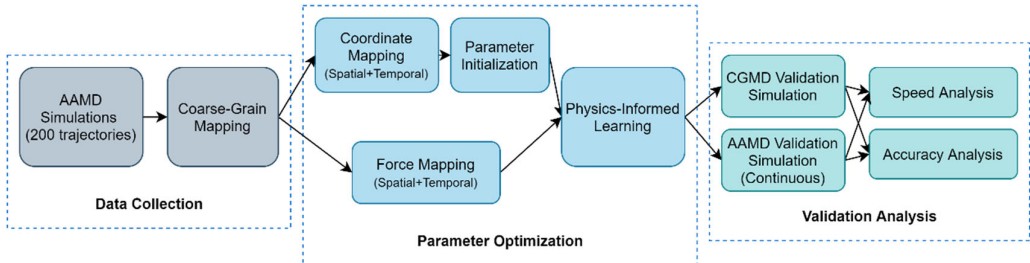

**Figure 2.** Illustration of the proposed CG modeling pipeline.

We converted our reference data to the CG scale to use the AAMD validation simulation data for training. This reference trajectory obtained in Appendix A.1. (Appendix A) was processed by mapping the extracted coordinate and force data to the CG scale both spatially and temporally. Spatial mapping was conducted by computing the center of mass and the sum of forces for each atom group, constituting a bead, according to:

$$X_{I,CG} = \frac{\sum_i w_i x_{i,AA}}{\sum_i w_i}, \tag{1}$$

$$F_{I,CG} = \sum_i f_{i,AA}, \tag{2}$$

where $X_{I,CG}$ and $F_{I,CG}$ represent the calculated position and force of bead $I$, $x_{i,AA}$ and $f_{i,AA}$ represent the position and force of atom $i$ within the atom group constituting bead $I$, and $w_i$ represents the mass of atom $i$ as a weighting factor. In addition to the spatial mapping, temporal averaging is performed to account for the greater temporal scales used in CGMD simulations. We averaged both coordinates and forces across the temporal dimension every 100 frames.

We initialized the parameters with traditional CG force field parameterization methods with bonded and nonbonded potentials. The bonded potentials are based on fixed lists of 2-, 3-, and 4-body interactions (bonds, angles, and dihedrals) modeled as spring harmonics with parameters as spring harmonic constants. The nonbonded potential is modeled with a Lennard-Jones (LJ) potential accounting for the weak dipole attraction between distant atoms and the hard-core repulsion between close atoms. The IBIM method is employed to initialize the new CG model force-field parameters, specifically the bonded parameters. Diverging from the original implementation, we incorporated the refinement of dihedral parameters in addition to the bonds and angles. From the ground truth, we extracted distribution functions $P(x)$ of variable $x$ representing the bond lengths, bond angles, or torsion angles. The potential function $U(x)$ is constructed using the Boltzmann relation:

$$U(x) = -k_B T ln P(x), \tag{3}$$

where $k_B$ is a parameter and $T$ represents the temperature. Furthermore, the bonded parameters can be modeled as harmonics:

$$U(x) = \frac{1}{2} k (x - x_0)^2, \tag{4}$$

where $x_0$ represents the respective equilibrium measurement and $k$ represents the harmonic constant. Thus, the Boltzmann inversion relationship between distribution functions and harmonic constants can be illustrated as follows:

$$\langle x^2 \rangle - \langle x^2 \rangle = \frac{k_B T}{2k}, \tag{5}$$

where the equilibrium measurement $x_0$ is equal to the average position $\langle x \rangle$. For a network of these bonded interactions, these bonds, angles, and dihedrals are not independent, and thus when parameters for each of them are derived individually using this Boltzmann inversion relationship, the stiffness of the structure may be overestimated. Hence, there is necessity in further optimization to better match the reference distributions.

The parameters for the non-bonded LJ potential are initialized and approximated by VMD and are based on the SASA calculations of the beads. Further detail into this procedure and its calculations are given in (A3) in Appendix A.

With the LJ potential, $U_{LJ}$, between pairs of beads (denoted by $i$ and $j$ subscripts) is defined as shown in Equation (6):

$$U_{LJ} = \epsilon_{ij} \left[ \left( \frac{R_{min_{ij}}}{r_{ij}} \right)^{12} - 2 \left( \frac{R_{min_{ij}}}{r_{ij}} \right)^{6} \right] \tag{6}$$

The relations between the $\epsilon_{ij}$ and $R_{min_{ij}}$ pair parameters with their respective trainable parameters for individual beads are defined below in Equations (7) and (8), respectively.

$$\epsilon_{ij} = \sqrt{\epsilon_i * \epsilon_j}, \tag{7}$$

$$R_{min_{ij}} = \frac{R_{min_i}}{2} + \frac{R_{min_j}}{2}. \tag{8}$$

### 2.3. Physics-Informed ML Model

A force-matching approach helps preserve thermodynamic consistency by minimizing the error between the instantaneous ground-truth forces and predicted forces [19,20,29,30]. Our PIML model defines its trainable parameters as the CG force field parameters. The CG coordinates serve as the input to the model, and the model further predicts the total potential energy of the system. All physically relevant invariances are thus preserved. Leveraging an automatic differentiation function, we take the negative gradient of this energy with respect to the input coordinates, and thus effectively obtain the instantaneous predicted forces. The task is thus to learn the parameters to minimize the error between these predicted forces and ground-truth forces in the loss function.

The model architecture, shown in Figure 3, is detailed further below. The model contains an initial featurization layer that converts the input coordinates to the pairwise distances, bond lengths, bond angles, and torsion angles, as displayed in Figure 3. The model uses two physics-informed layers, containing the trainable parameters, for the prediction of energy: one is the Harmonic layer comprised of bond, angle, and dihedral terms as bonded potentials; and the other is the Lennard-Jones layer.

Within the Harmonic layer, the trainable parameters include the harmonic constants, whereas, in the LJ layer, the trainable parameters are the bead strength $\epsilon_i$ and the minimum radius, $R_{min_i}$, for each unique bead $i$. There exist 471 and 40 trainable parameters that comprise the bonded and non-bonded interactions, respectively, in the physics-informed model. For the dihedral potentials, the periodic representation accounts for the periodicity of dihedrals, where the phase shift angle was adjusted to fit the equilibrium value as the potential minima. The resulting energy governing the CG force field can be calculated as:

$$U_{CG} = \sum_{bonds} k_b(r - r_0)^2 + \sum_{angles} k_a(\theta - \theta_0)^2 + \sum_{dihedrals} k_d(1 + cos(n\psi - \phi)) + \sum_{i<j}^{atoms} \epsilon_{ij} \left[ \left( \frac{R_{min_{ij}}}{r_{ij}} \right)^{12} - 2 \left( \frac{R_{min_{ij}}}{r_{ij}} \right)^{6} \right], \tag{9}$$

where $k_a$, $k_b$, and $k_d$ are spring factors, $r$ is bond distance, $\theta$ is bond angle, $\psi$ is torsion angle, and $\phi$ is defined as the torsion phase shift angle, which acts as an equilibrium angle in the periodical representation.

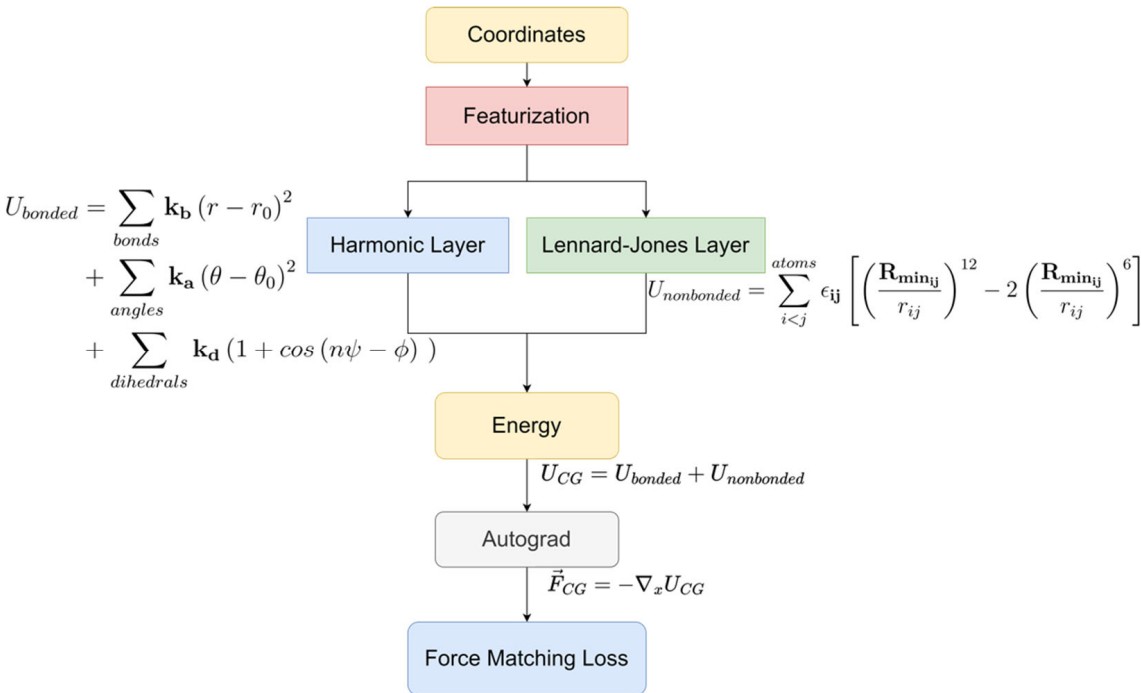

**Figure 3.** The proposed physics-informed model architecture.

The force $\vec{F}_{CG}$ can be calculated by the gradient of the potential

$$\vec{F}_{CG} = -\nabla_x U_{CG}. \tag{10}$$

with the loss function defined as

$$Loss = \left\langle \left( \vec{F}_{CG} + \nabla U_{CG} \right)^2 \right\rangle, \tag{11}$$

where $F_{CG}$ represents the predicted instantaneous force, and $U_{CG}$ represents the CG potential. This loss as a mean-squared error function between the predicted and the mapped ground-truth forces provides a means of minimizing their difference.

### 2.4. Validation and Verification

A simulation for CGMD validation is carried out using the learned parameters, together with a separate AAMD simulation, to measure the performance of our approach across the metrics of accuracy and speed. With regards to accuracy analysis, the RDFs are applied in providing insight into the distance distribution of particles around certain particles. The torsional analysis is applied in the form of free energy surface plots and the free energy was plotted along two dihedral quadruples, providing insight into the conformational states. From the plots, validation simulations are compared with the ground-truth training data using the dihedral pairs belonging to the S-protein receptor-binding domain (RBD) and S2 domain. Additionally, root-mean-square-deviation (RMSD) and root-mean-square-fluctuation (RMSF) are analyzed to monitor the structural stability of the compared models throughout their respective trajectories.

In addition to the simulated accuracy, we examined the speeds to measure our model's efficiency. The CGMD simulation was run for one microsecond and its simulation speed was carefully compared with the continuous AAMD validation simulation.

We extended the study to a solvated application beyond the solvent-free simulation environment. Using the same learned forces, we explicitly solvate the CG S-protein into a 18 nm × 18 nm × 18 nm MARTINI water box [31]. In this hybrid system, each MARTINI

water molecule is represented by a single bead (of mass 72 amu). To evaluate the accuracy of this solvated experiment, we ran an AAMD simulation of the S-protein solvated in a water box of TIP3 water molecules at the same 310K temperature in canonical (NVT) ensembles.

## 3. Results

With the learned parameters, the accuracy and speed of the CGMD simulations vs. the AAMD validation simulations are reported. Using the 97,905 coordinates and force frames, the parameter initialization for bonds, angles, and dihedrals, respectively, proceeded with 3 IBIM iterations. For each iteration, the trial simulations were conducted with 10 femtosecond timesteps, minimized for 500 picoseconds, and simulated for 4 ns. There exist 511 total learnable parameters that are learned with the physics-informed model configured with the Adam optimizer with a learning rate of 0.001 and a batch size of 256 for 10 epochs.

### 3.1. Accuracy Analysis

The CGMD and the AAMD simulations start from the same structure; the visualized protein structures of the starting and ending conformations after microsecond-level simulation in Figure 4 show their good alignment.

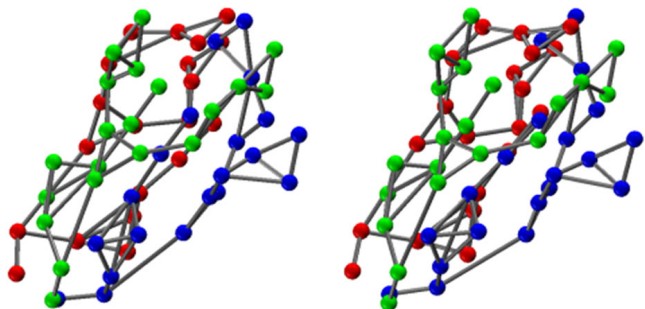

**Figure 4.** CG structure visualization: AAMD validation simulation final frame state (**left**). CGMD simulation final frame state (**right**).

The RDF measures the distribution of distances between the pairs of particles of two specified atom groups. For instance, Figure 5 defines these two groups to be some given "Atom #" and all "Atoms," respectively. Comparing the RDFs of our CGMD simulations with the ground-truth data, we measure the deviation between the mapped ground truth and the proposed CGMD simulations. As illustrated in Figures 5 and 6, the proposed PIML approach reproduces the structure in reference plots with reasonable accuracy, as it can capture the peaks in RDF.

To quantitatively measure the accuracy of the RDF plots, we incorporate Spearman's correlation coefficient [32] to measure the correlation between the CGMD and reference AAMD RDF plots. In Figures 5 and 6, the Spearman's correlation coefficients for each RDF plot are 0.7472, 0.6560, 0. 6278, 0.5690, and 0.9692 for atoms 7, 11, 14, and 19 and all atom pairs, respectively. This incorporation of a quantitative metric of Spearman's correlation coefficient confirms this reasonable correlation in Figure 5, and strong overall correlation in Figure 6.

Four representations are chosen in Figure 5: "Atom 7" and "Atom 14" plots present regions on the N-terminal domain (NTD), whereas the "Atom 11" plot references a bead located on the receptor-binding domain of the S-protein. "Atom 19" represents the base of the S2 subunit, closer to the stalk of the S-protein.

The free energy profiles are plotted as a function of dihedral angles. The plots are used to analyze and compare the torsion angles as a representation of the protein conformational states. Two separate pairs of torsional angles are displayed for such analysis: one is located on the receptor-binding domain, and the other is in the S2 subunit. Figure 7 shows that the proposed CGMD simulations match precisely the ground-truth training data. The

proposed physics-informed CG model captures the positions and peaks in the respective pairs with comparable accuracy to the AA model.

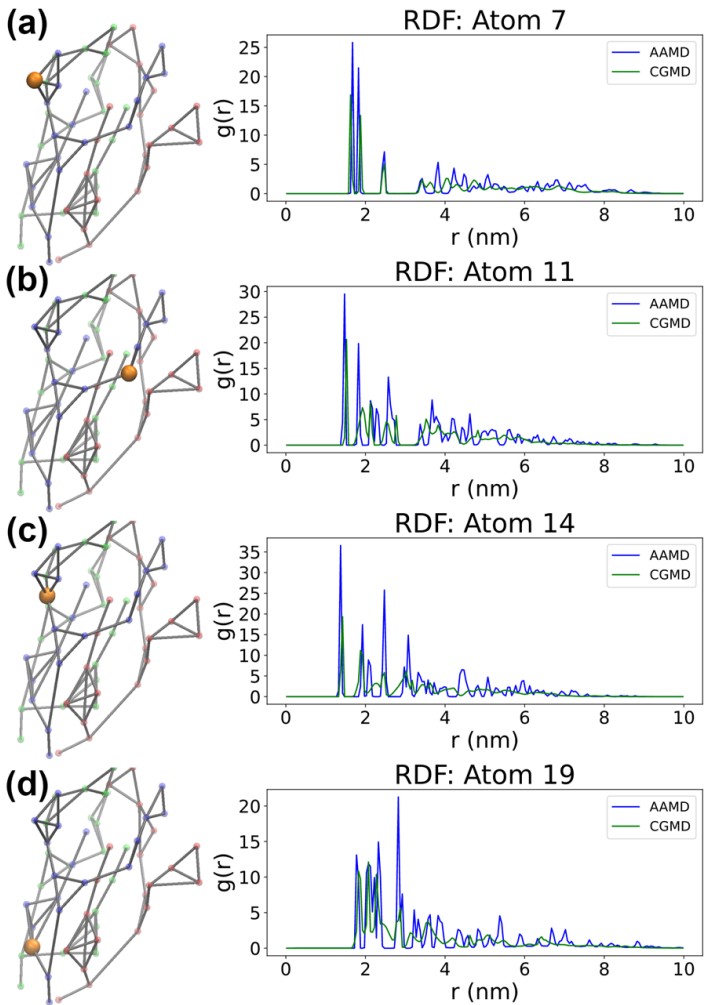

**Figure 5.** RDF plot from single reference atom comparison of the CGMD simulations vs. the AAMD validation simulations. In the colored beads visualizations: blue—chain A; red—chain B; green—chain C; orange—selected atoms. Spearman's correlation coefficients: (**a**) 0.7472; (**b**) 0.6560; (**c**) 0.6278; (**d**) 0.5690.

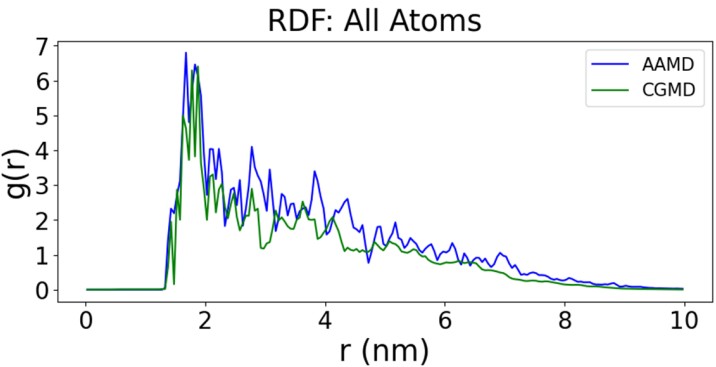

**Figure 6.** RDF plot of all atoms for comparison of CGMD vs. AAMD. Spearman's correlation coefficient: 0.9692.

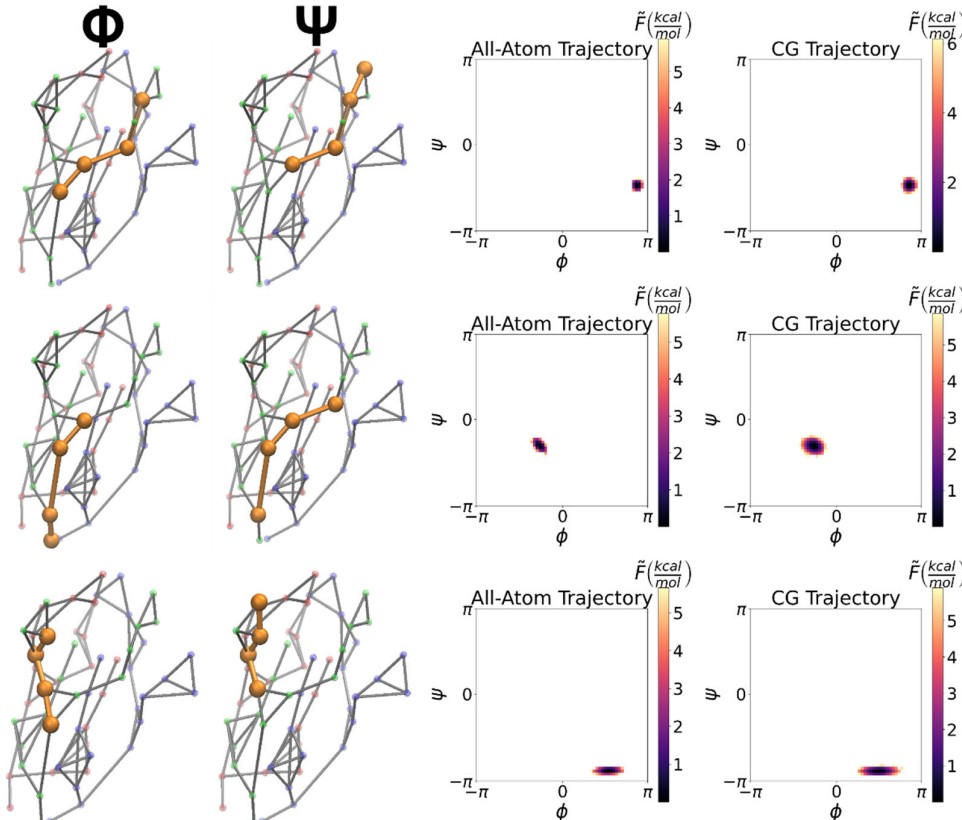

**Figure 7.** Free energy profiles of RBD pair (**top**), NTD pair (**middle**), and S2 subunit pair (**bottom**). Blue—chain A; red—chain B; green—chain C; orange—selected dihedral quadruplet.

Further analysis, showing the stability for the entirety of the microsecond, suggests the proposed physics-informed CG approach is feasible for long-term modeling of the SARS-CoV-2 S-protein. The evolution of the proposed physics-informed CG model trajectory was analyzed by calculating the RMSD values using the starting structure as a reference frame. The RMSD reveals the overall stability and conformational change of the whole protein. Protein coordinates are recorded every 10 picoseconds and the RMSD was calculated on the aligned trajectory. Figure 8 presents the RMSD of the proposed CGMD simulations along-side the AAMD validation simulations. The CGMD RMSD remains consistent throughout the full microsecond of simulation, indicating long-term structural stability.

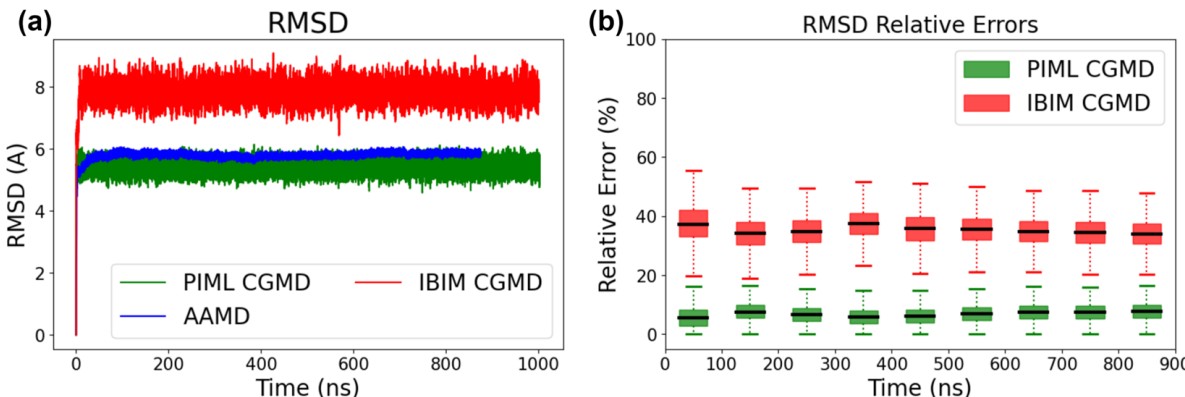

**Figure 8.** RMSD comparison between the proposed CGMD simulation and the AAMD validation simulation. RMSD in angstroms (**a**). RMSD relative errors (**b**).

The RMSD relative errors are included as well. Each error bar is normalized and extracted for statistics within a time period of 100 ns. All three simulations plotted below start with the same structure, and the relative error represents the relative error of our PIML and IBIM methods, respectively, with respect to the AAMD structures throughout their CGMD simulations. The calculation for such relative error is defined below:

$$e_t = \frac{|RMSD_{CG}(t) - RMSD_{AA}(t)|}{RMSD_{AA}(t)} \tag{12}$$

where $RMSD_{CG}(t)$ and $RMSD_{AA}(t)$ represent the RMSD of the CGMD and AAMD simulations, respectively, at time $t$.

The presented CGMD simulations appear to have greater fluctuations in comparison with the AAMD validation simulations, which indicates the CGMD is likely exploring a greater distribution of conformations. This is expected from the CG procedure, specifically how the averaging procedure smooths effective potentials, and thus how it facilitates enhanced sampling of the underlying phase space [33]. Our CGMD appears to have reached structures with RMSD values consistently closer to the RMSD values of the validation AAMD simulation compared with the IBIM approach. The animated trajectories of the AAMD validation simulation and the CGMD simulation are provided in the Supplementary Materials.

### 3.2. Speed Analysis

Both the AAMD validation simulations and the presented CGMD simulations are conducted on a local cluster, where each computing node consists of two Intel Xeon E5-2690v3 CPUs. By using the parallel NAMD package on 1 node with 24 CPU cores, the AAMD validation simulations with 1 femtosecond as the time step size produced 0.243 nanoseconds/day while the CGMD simulations with 10 femtoseconds as the time step size produced 9532.6 nanoseconds/day. This CGMD timestep was determined experimentally as the optimal speed that would maintain stable simulation. Specifically, we experimented with an array of timestep sizes ranging from 4 fs to 100 fs, and we settled on 10 fs for stability and speed. The experimental outcomes indicate that the presented CGMD validation simulations have a speed nearly 40,000 times faster than that of the AAMD validation simulations. Detailed measurements are presented in Table 2.

**Table 2.** Validation simulation comparisons using 24 CPU cores.

| Simulations | Time Step Size | Total Steps | Simulated Time | Simulating Time | Simulation Speed |
|---|---|---|---|---|---|
| AAMD | 1 fs | 100,000 | 0.1 ns | 35,557 s | 0.243 ns/day |
| CGMD | 10 fs | 500,000,000 | 5 μs | 45,318 s | 9532.6 ns/day |

### 3.3. Solvation Application

We assimilated our CG S-protein model with the MARTINI solvent using two separate cutoff configurations for nonbonded interactions. In this new configuration, the nonbonded interactions within the S-protein group are configured with a cutoff of 4.5 nm and smooth switching starting at 2.0 nm. Nonbonded interactions within the MARTINI solvent and between the solvent and the S-protein are configured with a cutoff of 1.2 nm and a smooth switching starting at 0.9 nm. For the RDF results as illustrated in Figure 9, our solvated model reproduces the structure in AAMD validation simulations collected in [14], resembling the significant peaks and retaining the overall structure of the protein.

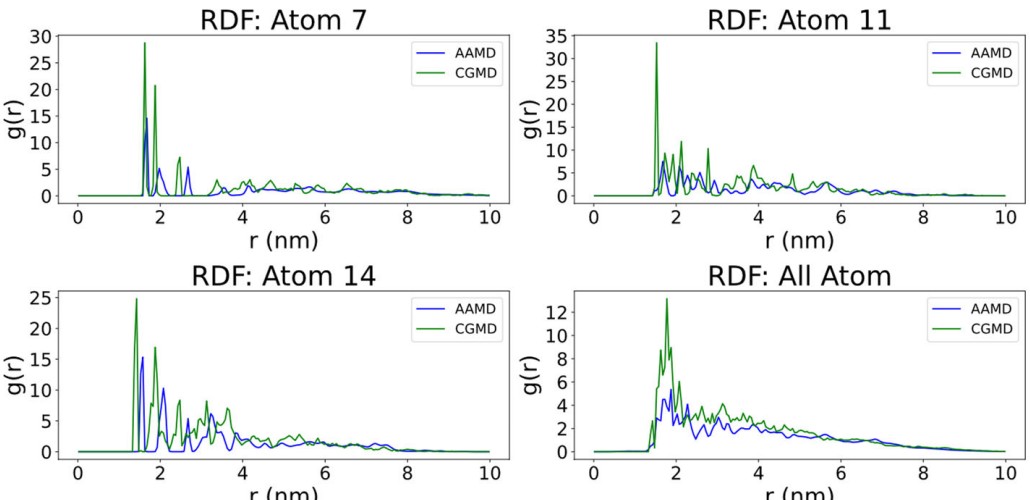

**Figure 9.** RDF plot from single reference atom comparison of the solvated CGMD simulations vs. the solvated AAMD validation simulations. "Atom 7" and "Atom 14" reflect on the NTD, and "Atom 11" reflects the receptor-binding domain.

## 4. Discussion

We presented an artificial intelligence-enabled model for multiscale CGMD simulations. The PIML approach to the model parameterization includes two phases: (1) using AAMD simulations to generate the ground truth for learning parameters and (2) using the learned parameters to run long-term CGMD simulations. The physics-informed bottom-up CGMD model simulations are compared with the ground truth AAMD simulations, the gold standard in accuracy, indicating a resemblance of the conformation. The proposed CG model is significantly faster than the AAMD simulation model. With the aggressive CG approach, the proposed model achieves nearly 40,000× the speed of the AAMD simulations.

The work underscores the following contributions toward more efficient multiscale modeling:

- The approach demonstrates the superiority of the supervised ML in deriving a CG model.
- In combining ML with molecular dynamics, our approach immensely accelerates simulations compared with the conventional AA models while maintaining stability and structural accuracy.
- The gained efficiency can elucidate protein mechanisms and render a great impact on future simulation studies by relieving the ongoing concerns about timeliness.

The application of our model into a solvated environment was presented and no term in our CG model was calibrated to reproduce the solvated reference. While rough structural accuracy was preserved, most clearly seen with the RDF plots, a limitation was noticed in our solvated simulation; the protein is observed to contract more than the reference. It is likely that calibration to the cutoffs, as well as the switching value, could yield better accuracy, and we intend to explore this as a future work. The proposed method underscores an important step forward in extending these large systems to actual applications in cases that it was not explicitly parametrized to reproduce, and in future works, we intend to adapt this proposed approach to binding of the S-protein with the ACE-2 receptor.

**Supplementary Materials:** The following supporting information can be downloaded at: https://www.mdpi.com/article/10.3390/computation11020024/s1.

**Author Contributions:** Conceptualization, methodology, software, validation, formal analysis, investigation, data curation, and visualization, D.L., Z.Z. and P.Z.; writing—original draft preparation, D.L.; writing—review and editing, D.L., Z.Z., P.Z., Y.D., M.R. and M.S.; supervision, project administration,

and funding acquisition, P.Z., Y.D., M.R. and M.S. All authors have read and agreed to the published version of the manuscript.

**Funding:** The project is sponsored by Stony Brook University's OVPR and IEDM COVID-19 Seed Grant, PIs: P.Z., Y.D., M.R. and M.S.

**Data Availability Statement:** Data generation was conducted through NAMD while machine learning and data analysis were conducted by our own code that is available upon request.

**Acknowledgments:** The project is supported by the SUNY-IBM Consortium Award, IPDyna: Intelligent Platelet Dynamics, FP00004096 (PI: Y.D., Co-I: P.Z.). All simulations were conducted on the AiMOS supercomputer at Rensselaer Polytechnic Institute through an IBM Faculty Award FP0002468 (PI: Y.D.) and the Seawulf cluster at Stony Brook University.

**Conflicts of Interest:** The authors declare no conflict of interest.

## Appendix A

### Appendix A.1. All-Atomic Simulations

To obtain the reference data, we first conducted AAMD simulations on the AiMOS supercomputer, a heterogeneous system architecture that includes IBM POWER9 CPUs connected to NVIDIA TESLA V100 GPUs, and the local computing cluster Seawulf at Stony Brook University. We utilized the CHARMM-36 force field [34] in describing the system in a vacuum canonical ensemble at 310K. Using NAMD software, conjugate gradient and line search energy minimization (10 picoseconds) was run prior to 400 picoseconds of simulation (1 fs timestep). From the stable simulation range, we randomly generated 200 different initial positions and orientations to branch off into separate, unique simulations. This was carried out to include replicas to address the chaotic component of MD simulations. From these simulations, frames containing coordinate and force data were collected every fs. A total of 9.7905 ns of the simulation data were accumulated, which upon mapping yielded 97,905 frames of coordinates and forces. From here on, this data constitutes our ground-truth data that represents the reference data the CG model aims to match.

### Appendix A.2. Dihedral Potential Term

For the dihedral potentials, they can be represented in two ways: quadratic representation of Equation (A1) and periodic representation of Equation (A2). The quadratic form represents the dihedral potential in the same manner as bonded potentials, where the trainable constants are analogous to spring constants. The periodic representation accounts for the periodicity of dihedrals, where the phase shift angle was adjusted to fit the equilibrium value as the potential minima.

$$U_{CG} = \sum_{bonds} k_b(r - r_0)^2 + \sum_{angles} k_a(\theta - \theta_0)^2 + \sum_{dihedrals} k_d(\psi - \phi)^2 + \sum_{i<j}^{atoms} \epsilon_{ij} \left[ \left( \frac{R_{min_{ij}}}{r_{ij}} \right)^{12} - 2 \left( \frac{R_{min_{ij}}}{r_{ij}} \right)^{6} \right], \quad \text{(A1)}$$

$$U_{CG} = \sum_{bonds} k_b(r - r_0)^2 + \sum_{angles} k_a(\theta - \theta_0)^2 + \sum_{dihedrals} k_d(1 + cos(n\psi - \phi)) + \sum_{i<j}^{atoms} \epsilon_{ij} \left[ \left( \frac{R_{min_{ij}}}{r_{ij}} \right)^{12} - 2 \left( \frac{R_{min_{ij}}}{r_{ij}} \right)^{6} \right], \quad \text{(A2)}$$

The torsion angle distribution can be plotted to depict the unimodality in Figure A1 and thus confirm the choice of n = 1 as the multiplicity for the periodic representations. The distributions in Figure A1 present more common conformations with the yellow color, where the means are the respective equilibrium states. While quadratic, or n = 0 representation also fits this unimodality, we understand that long-term secondary structural changes are unlikely to be modeled properly with this quadratic dihedral form. Thus, we favor the use of the periodic form, which lends itself to more flexibility in case of additional states.

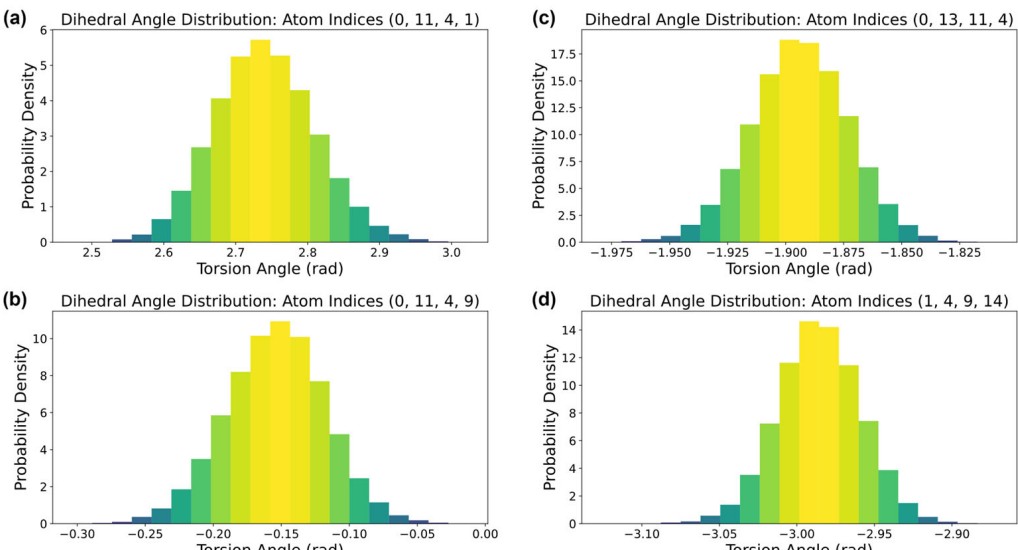

**Figure A1.** Randomly selected examples of torsion angle distributions for 4 dihedrals of atom indices of (**a**): (0, 11, 4, 1); (**b**): (0, 11, 4, 9); (**c**): (0, 13, 11, 14); and (**d**): (1, 4, 9, 14).

*Appendix A.3. Parameter Initialization*

The traditional IBIM used to initialize the bonded parameters process follows the following procedure: reference distributions extracted from ground-truth AAMD simulations. Initial bonded parameter "guesses" are obtained through the relation between references and bonded parameters in Equation (5). A trial simulation is run by configuring a short CG simulation with the aforementioned parameter guesses under the environment setup specified in Section 3. Distributions are extracted and compared with the reference AAMD distributions, and we then scale the bonded parameters accordingly to better match the distributions. This procedure of trial simulations and scaling parameters is iterated until the distributions match within reasonable tolerance. Our procedure involved 3 iterations until the parameters (denoting stiffness) extracted from its distributions are roughly within a 25% average deviation from that of reference [6]. Figure A2 illustrates the IBIM refinement of the parameters to initialize our parameters and match the reference distributions to reasonable accuracy after three iterations.

The nonbonded LJ parameter initialization based on SASA calculations is described as follows [29]. In this procedure, each bead $i$ was assigned an LJ strength $\epsilon_i$ based on:

$$\epsilon_i = \epsilon_{max} \left( \frac{SASA_i^{hphob}}{SASA_i^{tot}} \right)^2, \qquad (A3)$$

where $SASA_i^{hphob}$ and $SASA_i^{tot}$ represent the hydrophobic and total solvent-accessible surface areas of domain $i$, respectively, and $\epsilon_{max}$ is the user-controlled maximum energy for the LJ potential well depth. The reasoning behind using the SASA to determine $\epsilon_i$ is to allow hydrophobic beads to aggregate and hydrophilic beads to dissolve in the solvent, which is implicitly present in the CGMD simulations. The user-controlled $\epsilon_{max}$ was selected to be 20 kcal/mol based on approximations from findings in previous studies [29]. It is noted that while the user-defined constants are often tested for closest agreement with AAMD simulations in other studies [5], they will be later refined in the methodology as parameters by the ML model. The LJ potential radius $r_i$ (with the minimum of $R_{min_i}$) is given by the radius of gyration of the group of atoms constituting bead $i$, which is increased by a user-defined addition, e.g., an increment of 1 Å was selected in this work which accounts for the fact that each atom has a radius typically of 1–2 Å.

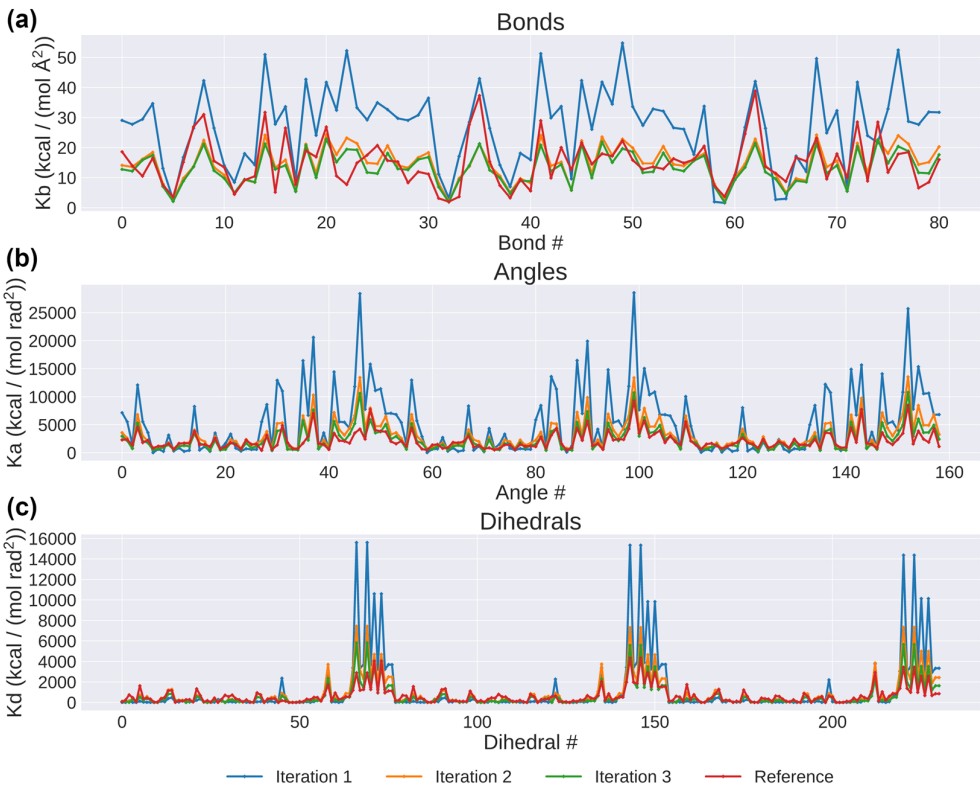

**Figure A2.** Illustration of IBI method's initialization of parameters for 3 iterations. Bonds (**a**); angles (**b**); dihedrals (**c**).

*Appendix A.4. Parameter Learning*

Figure A3 displays the loss plot over the training process. Both training and validation losses approached convergence after 4 epochs. The optimization of each individual bonded and non-bonded parameter over the 10 epochs is visualized in Figure A4. Hyperparameters of the network were determined experimentally to reach lower and faster convergence of the training loss. In our PIML, there exists two different groups of hyperparameters: the layers and trainable parameter count that were determined by the physics knowledge and the protein structure; and the learning rate, optimizer, learning rate decay scheduler, and batch size which were tuned experimentally. The range of learning rates we experimented with was 0.0005 to 0.003, and we settled on 0.001. The range of batch sizes experimented with was 16 to 512, and we settled on 256. The learning rate decay scheduler was experimented with along the full 10 epochs; the rate of decay ranged from 0.1 to 0.3, and we settled on 0.3.

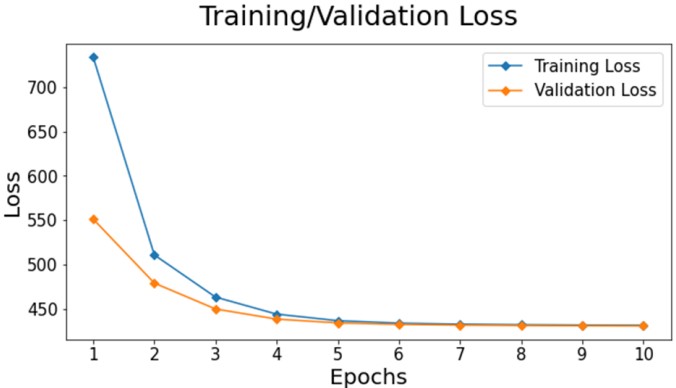

**Figure A3.** Training and validation loss vs. epochs.

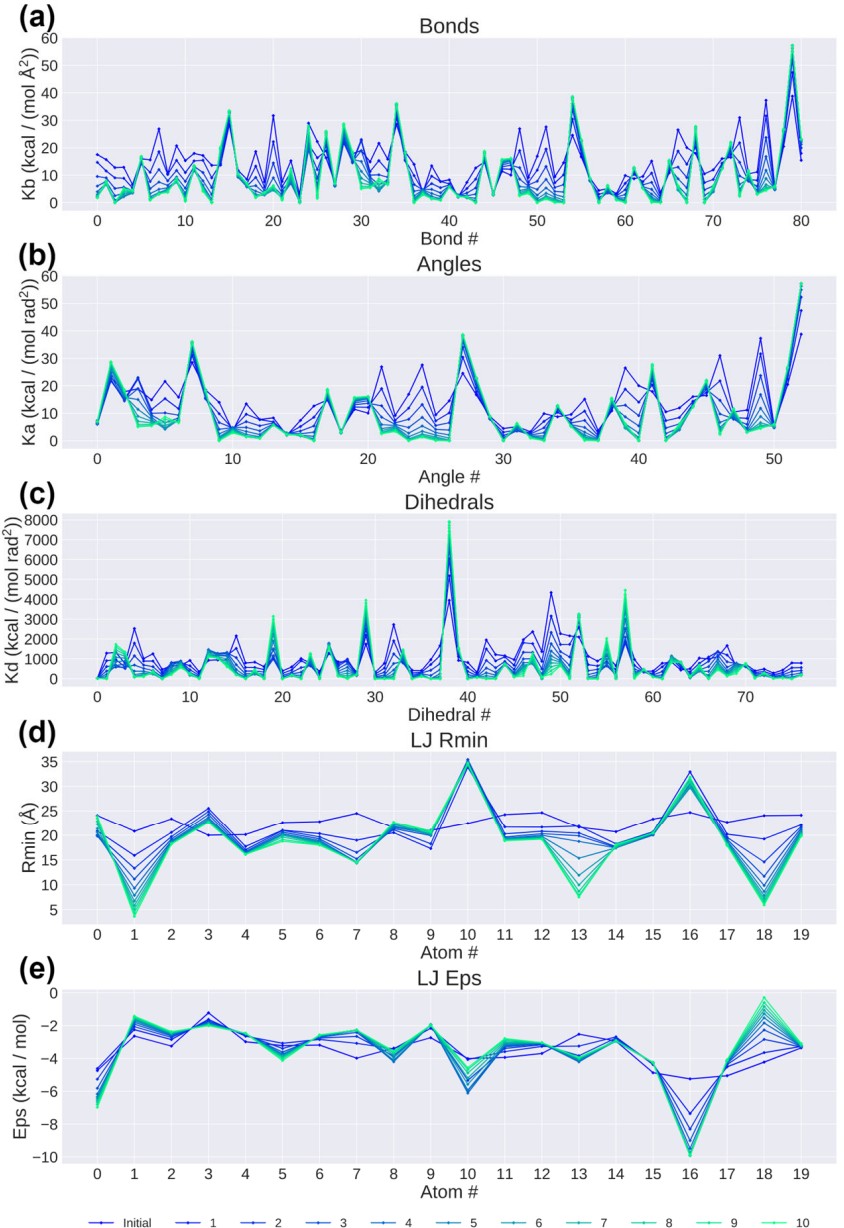

**Figure A4.** Physics-informed parameter learning for over 10 epochs. Bonds (**a**); angles (**b**); dihedrals (**c**); LJ Rmin (**d**); LJ Epsilon (**e**).

### Appendix A.5. ML Refinement on LJ Terms

We delved further into some specific changes reflected in the RDF measurements because of our ML design. The ML procedure indicates significant refinements in the model parameterization, particularly on the non-bonded LJ potential terms. Within this refinement is the very noticeable decrease in both the epsilon and the associated well-depth terms. Upon further investigation, it is shown that the model's calculated energies begin as positive (repulsion) and gradually become negative (attraction) by the end of the training, demonstrating the proper optimization to match the distances of the ground-truth data. In comparison with the IBIM trial results, specifically on the atom pair between atom numbers 17 and 46, the learned distances are more consistent with the ground-truth result, as shown in Figure A5. Furthermore, the incorporation of a quantitative metric of Spearman's correlation coefficient, which is 0.5831 for the PIML CGMD and −0.0376 for the IBIM CGMD, confirms this advantage.

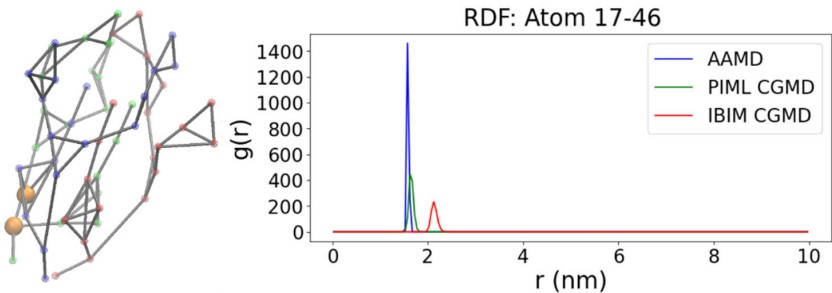

**Figure A5.** RDF plot between the atoms 17 and 46. Blue—chain A; red—chain B; green—chain C; orange—selected nonbonded atom pair. Spearman's correlation coefficients: (PIML CGMD) 0.5831; (IBIM CGMD) −0.0376.

*Appendix A.6. RMSFs for Bonded Interactions*

The comparison RMSFs of our CGMD simulation and the ground-truth AAMD simulations is shown in Figure A6. The results indicate that the PIML yielded a relatively accurate fit to the AAMD fluctuations. The difference between the ground-truth and continuous validation data in this case mainly stems from the temporal averaging in the ground-truth data, which may have dampened some fluctuations.

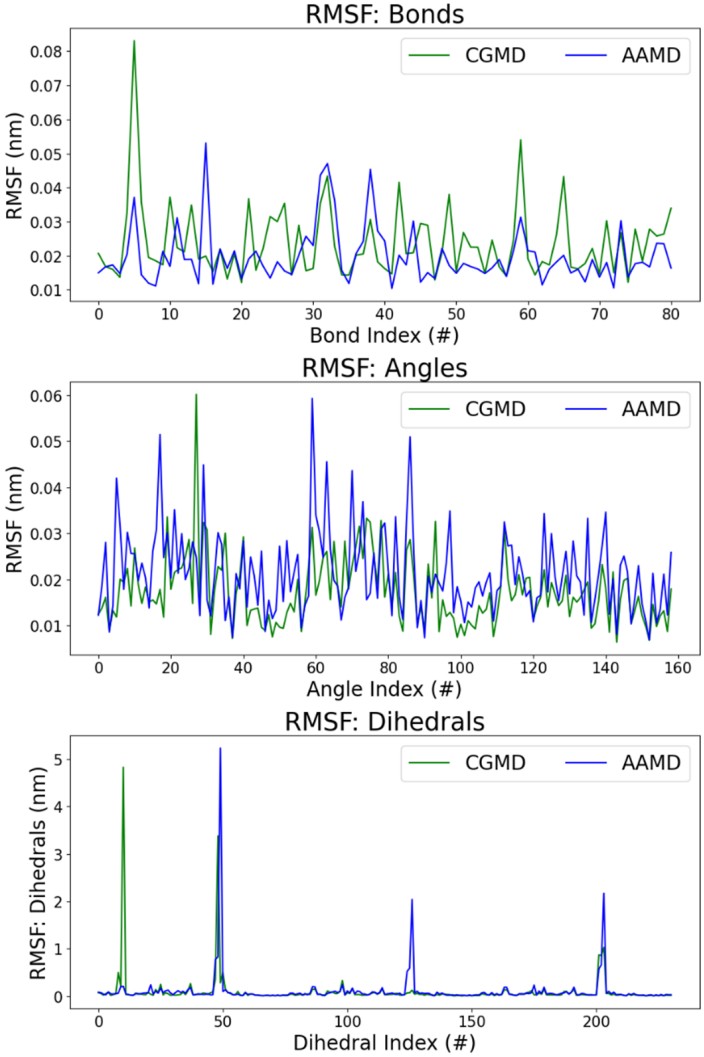

**Figure A6.** RMSF comparison between the proposed CGMD simulation and the AAMD validation simulations.

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
