# Peer review of "Coarse-Grained Modeling of the SARS-CoV-2 Spike Glycoprotein by Physics-Informed Machine Learning"

_computation, doi:10.3390/computation11020024_

Round 1

Reviewer 1 Report

A physics-informed machine learning framework (PIML) is proposed in this manuscript for parameterizing and optimizing coarse-grained molecular models. The author claims that a PIML-CGMD simulation of SARS-CoV-2 spike glycoprotein in a vacuum is shown to accelerate molecular dynamics simulation 40,000 times faster than conventional all-atom MD simulation. Technically, the approach and experimental setup used to validate the accuracy of the proposed model seem reasonable. However, the results and discussion need to be explained in more detail.

Major Concerns:

1. There is no discussion of other ML-based force fields in the manuscript, and the significance of their proposed model should be discussed.

2. What are the trial simulations the author mentions in the results? What is the simulation setup? Could you please give a brief explanation?

3. There is no detailed description of the RMSD relative error plot.

4. Is code and data available on GitHub or Zenodo?

Minor Concerns:

1. When represented as an all-atom version, what residue corresponds to Atoms 7,11,14, and 19? Would it be possible to include this information as a table in the supplementary material?

2. Which model was used to measure the related RMSD error for CGMDs? Are RMSD errors related to AAMD?

Author Response

Dear Editor and Reviewer-1:

Thank you for your encouraging remarks, and many constructive revision comments to which we furnish our responses on a point-by-point basis. We hope to hear good news from you soon.

Reviewer 2 Report

The authors of this manuscript propose a new coarse-grained MD (CGMD) based on machine-learning approach using all-atom MD simulation (AAMD) trajectories as training set. The present method is applied to SARS-CoV-2 glycoprotein. The authors present the better results compared with the other CCMD technique, IBIM method. An effective CGMD technique is desirable to handle the bigger protein systems in a practical sense and the present study is interesting and significant. Thus, the present study will be worth publishing after the following points are reconsidered.

1.       For general readers, the machine learning algorithm used in this manuscript should be simply explained.

2.       In Figures 5, 6 and 9, RDF plots are presented. The coincidence between the RDF plots of AAMD and CGMD looks not so good. The authors should show the quantitative measure and criterion of both plots, e.g., correlation coefficient. By the way, the authors should present the clear definition of RDF. Is it the average of RDFs between an atom in the protein and water oxygen?

3.       How are the four atoms for the calculation of dihedral angles selected in Figure 7? Furthermore, same calculation should be applied for other sets of four atoms in order to confirm that the results of AAMD and CGMD show the similar results.

Minor point;

In page 2, line 69, “(IBM)” should be “(IBIM)”.

Author Response

Dear Editor and Reviewer-2:

Thank you for your encouraging remarks, and many constructive revision comments to which we furnish our responses on a point-by-point basis. We hope to hear good news from you soon.

Reviewer 3 Report

The paper by Liang et al. presents a coarse-grained approach for the Sars-cov2 spike protein and compared it with the all-atomic MD simulation outcomes. Overall, I like the idea, however, things become messy in simulating CG model, and hard to validate and explore/obtain the drug-binding mechanism. I suggest acceptance with minor comments. 

I would suggest authors provide additional details on PIML framework for others to reproduce and employ this technique. Is this code open source? Please add a GitHub link as well. AA (45K atoms) reducing to 60 beads require a fair amount of discussion (topology preserving neural network), which authors have not addressed. Is the beads charged? How are electrostatic interactions gets modeled?

I was wondering about using 200 trajectories for coarse grain mapping. Please elaborate. 

Author Response

Dear Editor and Reviewer-3:

Thank you for your encouraging remarks, and many constructive revision comments to which we furnish our responses on a point-by-point basis. We hope to hear good news from you soon.

Reviewer 4 Report

January 13, 2023

Title: Coarse-grained modeling of the SARS-CoV-2 spike glycoprotein by physics-informed machine learning

Authors: David Liang, Ziji Zhang, Miriam Rafailovich, Marcia Simon, Yuefan Deng and Peng Zhang

Overview and general recommendation:

In this manuscript, the structural study of SARS-CoV-2 spike protein using a molecular dynamics simulation with a coarse-grained (CG) model was presented. Key to this work is the development of a “physics-informed machine learning (PIML)” framework for parameterizing CG force fields based on a force-matching scheme. They applied the framework to SARS-CoV-2 spike protein and validated the CG parameters against all-atom (AA) simulations. The presented CG simulations show better agreement with the referenced AA simulations compared to existing method, IBIM CG (e.g. Fig. 8). However, I have several concerns. First, the advantages of this framework are not clear, as there are similar force matching approaches using ML. Second, no data were provided to prove enhanced conformational sampling using the CG model (for example, Figure 7 shows very narrow distributions, so 1 microsecond CG is clearly not enough). Third, they parameterized only for the closed state of SARS-CoV-2 (PDBID: 6vxx), which severely limits the applicability of their CG model. Unfortunately, this work is not suitable for the publication at the current form. Some comments are following.

Comments:

(1)   The manuscript does not provide detailed descriptions of ML, such as the types and numbers of neurons and layers. To clarify the advantages of this work, the ML details including how they determined hyperparameters, must be given.

(2)   Isn't the referenced AA simulation too short? If my understanding is correct, only 10 ns of AA simulation data was used for training. I wonder if the training results in parameters that reproduce the local state, as the data presented suggests. What happens if you extend the CG simulation even longer (tens of microseconds)?

(3)   Why was the gas-phase AA simulation used as ground-truth? Wouldn’t it be better to train the parameters against AA simulations in water? This may be related to the rather poor performance of the solvated system (3.3. Solvation application). It is worth discussing the reason behind the presented framework choices.

(4)   What about the “noise problem” in the force matching approache?

(5)   How was the timestep of the presented CG model determined (estimated)?

(6)   It is worth discussing how this framework can be extend to a two-state model to analyze "open-close" motion of SARS-CoV-2 spike protein.

(7)   In SI, the movie for AA simulation has a large motion at the beginning, which the authors may want to discard for the publication.

Author Response

Dear Editor and Reviewer-4:

Thank you for your encouraging remarks, and many constructive revision comments to which we furnish our responses on a point-by-point basis. We hope to hear good news from you soon.

Round 2

Reviewer 4 Report

January 28, 2023

Title: Coarse-grained modeling of the SARS-CoV-2 spike glycoprotein by physics-informed machine learning

Authors: David Liang, Ziji Zhang, Miriam Rafailovich, Marcia Simon, Yuefan Deng and Peng Zhang

The revised manuscript has been largely improved and resolved my concerns. It turns out that the main focus here is how to handle large systems. To this end, the PIML framework in learning all interactions simultaneously and the integration of their model with Martini would contribute to the field and thus I suggest this work for publication.